# EquivAct: SIM(3)-Equivariant Visuomotor Policies beyond Rigid Object Manipulation

**Jingyun Yang**[1*]    **Congyue Deng**[1*]    **Jimmy Wu**[2]
**Rika Antonova**[1]    **Leonidas Guibas**[1]    **Jeannette Bohg**[1]

[1]Stanford University    [2]Princeton University

**Abstract:** If a robot masters folding a kitchen towel, we would also expect it to master folding a beach towel. However, existing works for policy learning that rely on data set augmentations are still limited in achieving this level of generalization. We propose *EquivAct* which utilizes SIM(3)-equivariant network structures that guarantee out-of-distribution generalization across object translations, 3D rotations, and scales by construction. Our method first pre-trains a SIM(3)-equivariant visual representation on simulated scene point clouds and then learns a SIM(3)-equivariant visuomotor policy on top of the pre-trained visual representation using a small amount of source task demonstrations. In both simulation and real robot experiments, we show that the learned policy directly transfers to objects that substantially differ in scale, position, and orientation from the source demonstrations. Website: https://equivact.github.io

**Keywords:** Generalization in robot learning, domain adaptation, equivariance

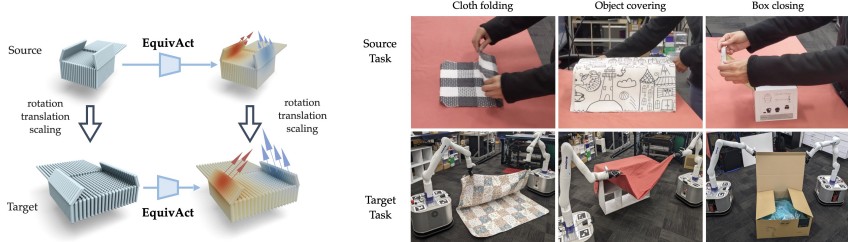

Figure 1: Our method takes a few example trajectories, demonstrated in a single source manipulation scenario, and generalizes zero-shot to scenarios with unseen object visual appearances, scales, and poses.

## 1    Introduction

Given a few examples of how to solve a manipulation task, humans can extrapolate and learn to solve variations of the same task where the objects involved have different visual or physical properties. Existing works in robot learning still require extensive data augmentation to make the learned policies generalize to varied object scales, orientations, and visual appearances [1, 2, 3, 4, 5]. Even then, augmentations do not guarantee generalization to unseen variations.

In this work, we focus on the problem of learning a visuomotor policy that can take a few example trajectories from a single source manipulation scenario as input and generalize zero-shot to scenarios with unseen object visual appearances, scales, and poses. In particular, we want the learned policies to be capable beyond mere pick-and-place of rigid objects and also handle deformable and articulated objects. Our insight is to *add equivariance into both the visual object representation and policy architecture*, so that the learned policy generalizes to novel object positions, orientations, and scales *by construction*. Although prior works have studied equivariant features for robot manipulation [6, 7, 8, 9, 10], most of these works handle only pick-and-place-like tasks and are based on open-loop policies that do not support sensory feedback during manipulation.

7th Conference on Robot Learning (CoRL 2023), Atlanta, USA.

We present *EquivAct*, a novel visuomotor policy learning method that can learn closed-loop policies for 3D robot manipulation tasks from demonstrations in a single source manipulation scenario and generalize zero-shot to unseen scenarios. Our method is composed of two phases: a representation learning phase and a policy learning phase. In the representation learning phase, the agent is given a set of simulated point clouds that are recorded from objects of the same category as the objects in the target task but with a randomized non-uniform scaling. While the proposed architecture is equivariant to uniform scaling, we need to augment the training data in this way to account for non-uniform scaling. With the simulated data, we train a SIM(3)-equivariant [11] encoder-decoder architecture that takes the scene point cloud as input and outputs the global and local features of the input point cloud. In the policy learning phase, we assume access to a small set of demonstrated trajectories of the task. With the demonstration data, we train a closed-loop policy that takes a partial point cloud of the scene as input, uses the pre-trained encoder from the representation learning phase to obtain the global and local features of the input point cloud, and then passes the features through a SIM(3)-equivariant action prediction network to predict end-effector movements.

We evaluate our method in three challenging tasks that go beyond the typical rigid object manipulation tasks of prior work [6, 7, 8, 9, 10]: cloth folding, object covering, and box closing (see Fig. 1). We show that our approach can successfully learn a closed-loop robot manipulation policy from the source manipulation demonstrations, and complete the target task zero-shot without any further adaptation.

## 2 Method

### 2.1 Preliminaries

**SIM(3)-equivariance.** Given a function $f$ which takes a point cloud $\mathbf{X} \in \mathbb{R}^{N \times 3}$ as input, we say it is SIM(3)-equivariant if for any 3D rigid transformation $\mathbf{T} = (\mathbf{R}, \mathbf{t}, s) \in \mathrm{SIM}(3)$ with rotation $\mathbf{R}$, translation $\mathbf{t}$, and scale $s$, the output of $f$ transforms coherently with the input, that is $f(\mathbf{TX}) = \mathbf{T}f(\mathbf{X})$. For the least trivial rotation component $\mathbf{R}$ of $\mathbf{T}$, [12] introduces a framework called Vector Neurons (VN)

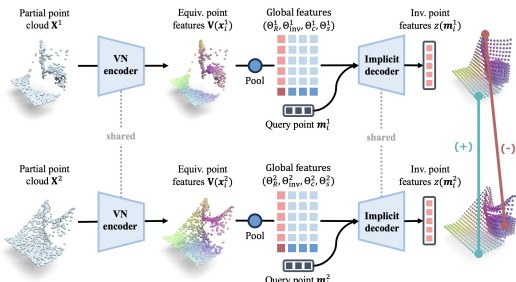

Figure 2: **Representation learning pipeline.**

for constructing rotation-equivariant point cloud networks. With the vector neuron building block, [11] then builds a SIM(3)-equivariant shape encoding using implicit representations. Given a partial point cloud $\mathbf{X}$, a VN encoder $\Phi$ encodes the input into a latent code $\Theta = \Phi(\mathbf{X})$ comprised of four components $\Theta = (\Theta_R, \Theta_{\mathrm{inv}}, \Theta_c, \Theta_s)$. Here, $\Theta_R \in \mathbb{R}^{C \times 3}$ is a rotation equivariant latent representation, $\Theta_{\mathrm{inv}} \in \mathbb{R}^C$ is an invariant latent representation, scalar $\Theta_s \in \mathbb{R}$ represents object scale, and 3D vector $\Theta_c \in \mathbb{R}^3$ represents the object centroid. More details of the encoder network can be found in [11]. The latent code can be decoded by an implicit decoder $\Psi(\mathbf{x}; \Theta)$ that takes a query position $\mathbf{x} \in \mathbb{R}^3$ as input and outputs a per-point feature $z(\mathbf{x})$.

**Problem formulation.** An agent is given a set of demonstrations $\mathcal{D}_{\mathrm{demo}} = \{\tau_n\}_{n=1}^{N_{\mathrm{demo}}}$. Each demonstration $\tau_i$ consists of a sequence of transition tuples $\{(o_t, a_t)\}_{t=1}^{T}$, where $o_t = (\mathbf{X}_t, \{\mathbf{y}_t^e\}_{e=1}^{E})$ is an observation that includes a point cloud of the scene $\mathbf{X}_t$ and the poses of all end-effectors $\{\mathbf{y}_t^e\}_{e=1}^{E}$, and $a_t$ is an action that consists of velocity and gripper commands for all end-effectors. We train a policy $\pi(a|o)$ from the demonstrations to take an observation as input and output an action $a$. We then deploy the learned policy in a setup with an unseen object out of the distribution of objects in $\mathcal{D}_{\mathrm{demo}}$ and is evaluated on the final reward of completing that target task.

### 2.2 Learning SIM(3)-equivariant 3D Visual Representations from Simulation Data

The first step of our method involves training a 3D representation for the objects involved in the manipulation task (illustrated in Fig. 2). We assume access to a dataset $\mathcal{D} = \{(\mathbf{X}^i \in \mathbb{R}^{N \times 3}, \mathbf{M}^i \in \mathbb{R}^{M \times 3})\}_{i=1}^{L_{\mathcal{D}}}$ containing point clouds of simulated scenes that include the objects of interest. In each

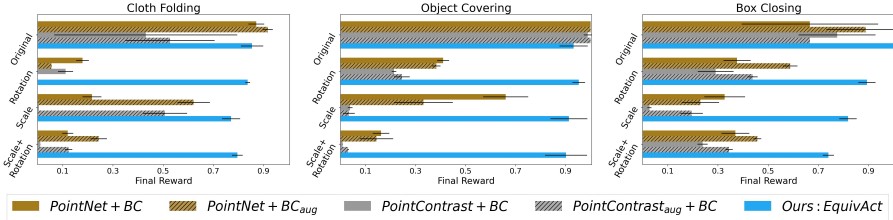

Figure 3: **Results for simulation experiment.** In all three manipulation tasks, our method outperforms prior methods that rely on augmentations to achieve generalization or utilize non-equivariant visual representations.

sample, we record the partial point cloud of the scene unprojected from one camera view $\mathbf{X}^i$ and the ground-truth mesh of the objects in the scene $\mathbf{M}^i$. We train the visual representation using a contrastive loss similar to [13]. More specifically, we sample a simulated data pair of object point clouds $(\mathbf{X}^1, \mathbf{M}^1)$ and $(\mathbf{X}^2, \mathbf{M}^2)$ where two different object instances share a similar pose, articulation, or deformation. Then, we aim to learn per-point features $\{z(\mathbf{m}_i^1)\}_{i=1}^M$ and $\{z(\mathbf{m}_i^2)\}_{i=1}^M$ at ground-truth mesh points. Since we know the two ground-truth meshes have one-to-one correspondences, we know that $z(\mathbf{m}_i^1)$ and $z(\mathbf{m}_i^2)$ correspond to the same points on the mesh and should have similar latent features. We then apply a PointInfoNCE loss [13] to train the latent representation.

## 2.3 Learning Generalizable Visuomotor Policies with SIM(3)-equivariant Architecture

After the representation learning phase, we freeze the encoder $\Phi$ and learn an action prediction head on top of the learned feature representation with a small number of human demonstrations.

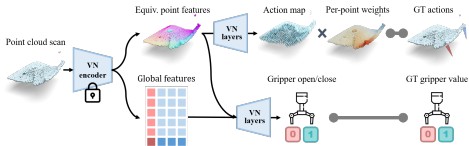

Figure 4: **Policy learning architecture.**

The policy learning architecture is illustrated in Fig. 4. Given input observation $o = (\mathbf{X}, \{\mathbf{y}^e\}_{e=1}^E)$, our policy $\pi$ first passes the input point cloud through the encoder $\Phi$ to obtain global feature $\Theta_R$ and per-point features $\{\mathbf{V}(\mathbf{x}_i)\}_{i=1}^N$ of the input scene point cloud. Then, the policy architecture is split into two branches. The offset prediction branch takes the local features as input and passes them through 3 VN layers to produce a velocity map $\{\hat{\mathbf{v}}_i\}_{i=1}^N$, which corresponds to the predicted velocity of the end-effector if the end-effector is close to the point location. The gripper action prediction branch takes the global feature as input and passes it through 3 VN layers different from the ones in the offset prediction branch to produce a binary gripper open/close prediction for each end-effector $\{\hat{g}_e\}_{e=1}^E$. During inference, we find for each end-effector $e \in \{1, \ldots, E\}$ a point $\mathbf{x}_{k_e}$ on the point cloud that is closest to the end-effector position $\mathbf{y}^e$ and then use the velocity map output of that position $\hat{\mathbf{v}}_{k_e}$ as the predicted velocity of the corresponding end-effector. The predicted end-effector velocity and gripper action predictions are then concatenated to form the output action $\hat{a} = (\{\hat{\mathbf{v}}_{k_e}\}_{e=1}^E, \{\hat{g}_e\}_{e=1}^E)$. We use a multi-part loss function that supervises the velocity predictions and the gripper control predictions of the policy separately. can be found in the appendix.

## 3 Experiments

### 3.1 Comparisons

We compare our method with several baselines, as described below. (1) **PointNet+BC:** a baseline behavior cloning (BC) algorithm that trains a neural network architecture, which takes a partial point cloud of the scene as input, encodes the input with a PointNet [14], concatenates the point cloud feature with robot proprioception, and then passes the concatenated vector through an MLP to predict robot actions. Since this baseline does not have a representation learning phase, it only uses the source task demos and does not utilize simulated data. (2) **PointNet+BC$_{aug}$:** a variation of *PointNet+BC* that augments the demonstration data on object pose and scale, aiming to make

test-time scenarios 'in distribution' with respect to the augmented data. The total amount of data used after augmentation equals the amount of data used in our method. (3) **PointContrast+BC:** this baseline first trains a latent representation of the 3D scene with PointContrast [13], and then learns an MLP that takes the learned features and robot proprioception as input to predict robot actions.

### 3.2 Simulation Experiments

**Tasks.** We evaluate our method on three robot manipulation tasks involving deformable and articulated objects: cloth folding, object covering, and box closing. More details about these tasks can be found in the appendix.

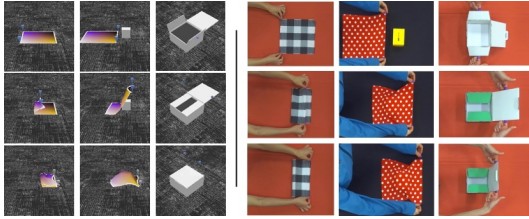

Figure 5: **Demonstrations for simulation (left) and real robot (right) experiments.**

**Evaluation.** We test our policy in four different setups. The *ID* setup places and sizes the object the same as the demonstrated scenario; the *Rot* setup randomizes the rotation of the object while keeping the object scale constant; the *Scale* setup randomizes the non-uniform scale of the object between $1\times$ and $2\times$ while keeping the change of aspect ratio within $1 : 1.33$ (shortest vs. longest dimensions) and keeping the object in a canonical pose; while the *Rot+Scale* setup randomizes both the rotation and non-uniform scaling of the object. Note that except for the *ID* setup, all other setups test the out-of-distribution performance of the trained policy. We measure the performance of a task using a task-specific reward function scaled between 0 to 1. See the appendix for more details.

**Results.** We present the simulation experiment results in Fig. 3. In the *ID* evaluation setup, all methods exhibit good performance. In the other three setups, the two **PointNet+BC** baselines display a significant performance drop. Even after adding augmentations in **PointNet+BC+Aug**, the baseline is still not able to recover the in-distribution performance for any of the three tasks. Similarly, the **PointContrast+BC** baseline performs well in the *Rot* setup but struggles to achieve good performance in the evaluation setups that involve scaling. Our method, in comparison, could achieve good performance in all out-of-distribution setups, displaying a minor performance drop compared to the *ID* setup.

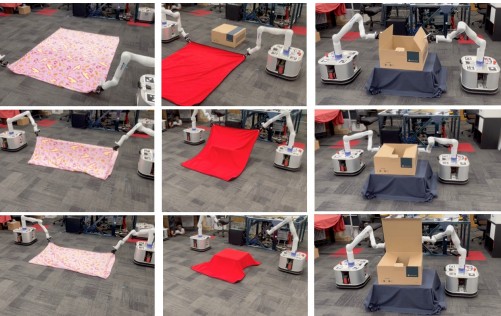

Figure 6: **Results of deploying the policy on hardware.** We deploy our learned policies to a mobile robot platform to manipulate target objects that are at least 6 times larger than the demonstrated objects. Even with such a large difference between source and target domains, our policy can zero-shot generalize to the target objects and can successfully finish the task.

### 3.3 Real Robot Experiments

To illustrate the effectiveness of our method in the real world, we qualitatively test it in a mobile manipulation setup. More details can be found in the appendix and our website. In all three tasks, our method extrapolates from tabletop-scale demonstrations and successfully manipulates objects that are up to $6\times$ larger in size. We further demonstrate the robustness of our method by varying the position, rotation, scale, and appearance of the objects in the task (see Fig. 6). Our qualitative evaluations show that our method can robustly generalize to all these variations of the task.

## 4 Conclusion and Future Work

We presented *EquivAct*, a visuomotor policy learning method that learns generalizable close-loop policies for 3D manipulation tasks. We showed that our method successfully generalizes from a small set of source demonstrations to a diverse set of different target scenarios in both simulation and real robot experiments. Extending our method to use large demonstration datasets to build a versatile multi-task equivariant policy is a promising direction for future work.

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

# 5 Appendix

## 5.1 Visualization of Learned Visual Representations

In Figure 7, we visualize the per-point features learned by the first phase of our method. In the figure, the encoder features are equivariant vector-valued features on the partial point cloud observations, and the visualizations are done on their invariant components (channel-wise 2-norms). The decoder features are invariant scalar-valued features on the complete objects. The RGB values are computed via PCA within each task. All point clouds are aligned to the canonical pose for visualization.

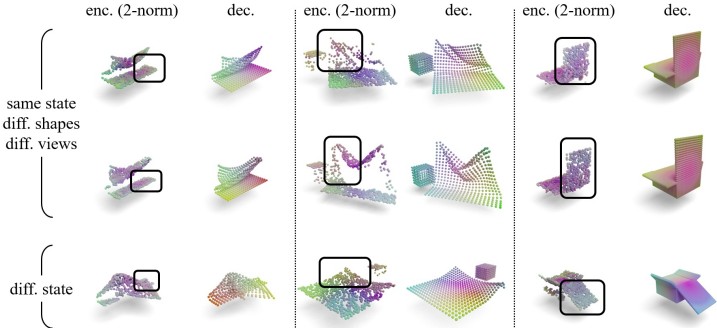

Figure 7: **Visualizations of per-point features.** *Top two rows:* Objects of different shapes viewed from different camera angles but at the same poses. Both encoder and decoder features show strong correspondences within each state due to the contrastive learning. *Bottom row:* Objects from a different state. The features become different from the above rows.

## 5.2 Training Details

**Loss function of the policy learning phase.** We use a loss function that consists of two parts for training the policy. To supervise the outputs of the velocity map, we want to apply a loss term on every point on the point cloud. However, points that are located farther away from any end-effector in the demonstration should receive less supervision than points closer to one of the end-effectors. To achieve this, we weight the loss term applied to each point on the point cloud such that the closer a point is to the end-effector, the higher the loss is weighted. We construct a velocity prediction loss defined by $\mathcal{L}_{\text{vel}} = \frac{1}{N} \sum_{i=1}^{N} \sum_{e=1}^{E} w(\mathbf{x}_i, \mathbf{y}_e) \cdot (\mathbf{v}_i - \hat{\mathbf{v}}_i)$, where $\mathbf{v}_i$ denotes the ground truth velocity of the $i$-th end-effector, $\hat{\mathbf{v}}_i$ denotes the predicted velocity of the $i$-th end-effector, and $w(\mathbf{x}_i, \mathbf{y}_e) = \exp(-(\mathbf{x}_i - \mathbf{y}_e)^2/(2\sigma))$ is a weighting function that is large when $\mathbf{x}_i$ is close to $\mathbf{y}_e$ and small when $\mathbf{x}_i$ is far from $\mathbf{y}_e$. The gripper control prediction loss is simply defined as the MSE loss between the predicted gripper actions $\hat{g}_e$ and the ground truth actions $g_e$: $\mathcal{L}_{\text{grip}} = \text{MSE}(\hat{g}_e, g_e)$. The policy learning loss is a weighted sum of the velocity prediction and gripper loss terms: $\mathcal{L} = \lambda_{\text{vel}} \mathcal{L}_{\text{vel}} + \mathcal{L}_{\text{grip}}$, where $\lambda_{\text{vel}}$ is a weighting term.

## 5.3 Task Details

We evaluate our method in three manipulation tasks in both simulation and real robot experiments: (1) *Cloth Folding:* two robots fold a piece of cloth together by grasping two corners of the cloth. This task is designed to test if our method can handle deformable object manipulation. (2) *Object Covering:* two robots grasp a cloth on two corners and place the cloth on top of an object so that the object is fully covered. This task is designed to test handling scenarios with several objects. (3) *Box Closing:* two robots close a box with three flaps by first closing the side flaps and then closing the larger front/back flap. This task tests manipulation with articulated objects. In particular, in the simulated *Box Closing* task, we add a further challenge to the trained policy by removing the contents in the box at test time, so that the test-time objects have different shapes compared to the training-time object.

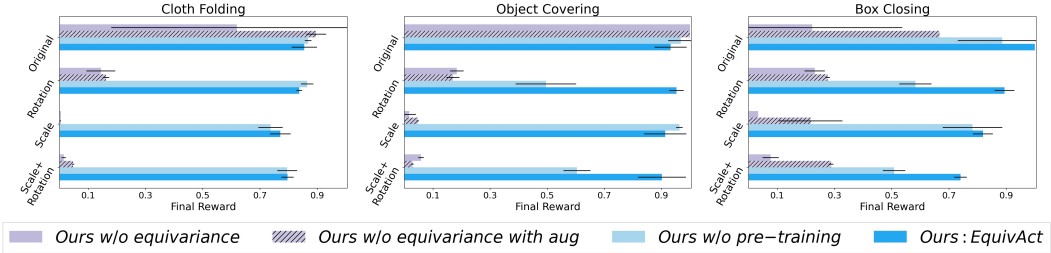

Figure 8: **Ablation results in simulation.** The comparisons show that the pre-training and the equivariance components of our framework positively contribute to the performance of our framework.

**Evaluation.** We measure the performance of a task using a task-specific reward function scaled between 0 to 1. In the *Cloth Folding* task, the reward is measured by how close the bottom two corners of the cloth are to the top two corners of the cloth; in the *Object Covering* task, the reward equals the percentage of the object volume within the convex hull of the final cloth mesh; in the *Box Closing* task, the reward equals the average percentage that each of the three flaps of the box is closed.

## 5.4 Details on Simulation Data

**Representation learning data.** Our simulation environments use PyBullet simulation engine [15]. For simplicity, we use floating end-effectors, which can physically push objects and can also attach to objects to imitate grasping. We generate randomized simulation data in each task to be used for representation pre-training. In *Cloth Folding* and *Object Covering* tasks, we generate representation learning data by grasping two corners of the object and performing a scripted pick and place motion to random target positions around the cloth. We collected 200 episodes of simulation data in each of the two tasks, varying the non-uniform scaling and camera viewpoint of each episode. Note that these 200 episodes are not successful executions of the target task and do not include recorded robot actions. In the *Box Closing* task, we generate representation learning data by taking snapshots of the object at various articulated poses. We collect a total of 20k images, equivalent to 200 episodes of episode rollouts in this task. We vary non-uniform scaling and camera viewpoint of each collected image.

**Demonstration data.** We collect 50 noisy demonstrations for all tasks. In each task, the 50 demos include the same object at the canonical pose. This means that the policy is trained using demonstrations of only one manipulation scenario.

## 5.5 Simulation Ablation Experiments

To illustrate that each component of our method individuallly contributes to the overall performance of our method, we compare our method with the following variations of our method: (1) **PointContrast_aug+BC:** a variation of *PointContrast+BC* that uses augmented object poses during feature training, which is a way of approximating equivariant features with Siamese training [16]. (2) **Ours w/o pre-training:** this ablation is a variation of our method without a pre-training phase, which trains the whole architecture, including the visual encoder and the action prediction head, together with the policy. (3) **Ours w/o equivariance:** this ablation uses a variation of our architecture that replaces all SIM(3) equivariant architecture with non-equivariant PointNet-based architecture with similar size. (4) **Ours w/o equivariance with aug:** this ablation is our method without SIM(3) equivariance, but with using augmented object poses during feature training.

Our ablation experiments are presented in Fig. 8. The performance of the **Ours w/o Pretraining** and **Ours w/o Equivariance** ablations show that having pre-training and equivariance in our framework positively contribute to the performance of our framework.

## 5.6 Real Robot setup

**Tasks and data collection.** We run our method in the same three manipulation tasks as our simulation experiments. In each task, our method takes 20 human demonstrations collected on a tabletop setup as training data. In the *Cloth Folding* task, we use a square cloth with a side length of 27.5cm; in the *Object Covering* task, we use a square cloth with a side length of 27.5cm to cover a box with size $13 \times 8 \times 8$cm; in the *Box Closing* task, we use a small $15 \times 13.5 \times 9.5$cm electronics box with three flaps. We use a ZED 2 stereo camera positioned above the table to record the movement of the objects and human hands. After data collection, we segment out the part of the point cloud of that corresponds to the objects the human is manipulating, and also parse out the human finger positions in each frame. We treat segmentation as a separate research problem, which could be addressed by state-of-the-art methods, such as [17]. However, to avoid conflating the evaluation of the performance of various segmentation methods with the main evaluation of our method, we segment out the objects using simple (robust) color filtering techniques.

**Mobile robot setup.** We train our method on data extracted from human demonstrations and then deploy it in the real world. The real robots operate in a large workspace with size $4 \times 3$ meters. We use holonomic mobile bases with a powered-caster drive system [18], and Kinova Gen3 7-DoF arms equipped with Robotiq 2F-85 grippers. We use Zed 2 stereo cameras positioned on the ceiling to obtain point clouds of the scene, then segment out the relevant parts of the scene. The policy takes the partial point cloud of the objects in the scene as well as the proprioception readings of the two mobile robots as input, then outputs a velocity and gripper open/close commands for each robot. For simplicity, we initialize the robot end-effectors close to the object and select an appropriate end-effector rotation for completing the task. As before, we treat segmentation as a problem beyond the scope of this work and use simple color filtering to separate the objects from the background and robots.

## 5.7 Evaluations on Task Variations

To illustrate the generalization capability of our method, we test it on various objects and initial poses. The first two columns show *Box Closing* policy on two differently sized boxes placed in different initial rotations. The same policy successfully closes the boxes in both cases. In the third column, we show that the *Cloth Folding* policy could handle different cloths that have distinct appearance, scale, and physical properties compared to the pink cloth blanket shown in Fig. 6.

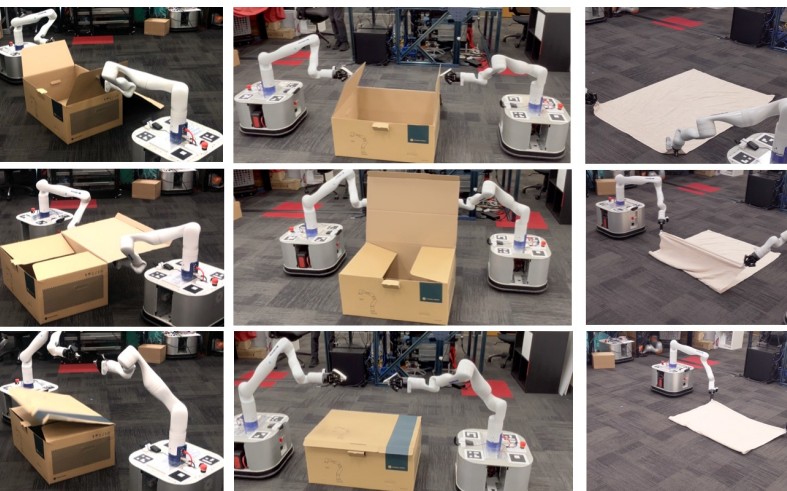

Figure 9: **Task variations.**

### 5.8 More Related Works

#### 5.8.1 Equivariant 3D Learning

A number of 3D deep learning works have studied the construction of neural network architectures that are equivariant to 3D transformations such as rigid transformations [19, 20, 21, 12, 22, 23, 24, 25], multi-part motions [26, 27, 11, 28], or more general and arbitrary group actions [29, 30, 31, 32, 33]. Such network designs enable natural and robust generalizations to out-of-distribution inputs by construction without additional data augmentation in the training process. In our work, we leverage the implicit shape representation from [11] powered by the network structures from [12] to facilitate more geometrically interpretable and generalizable feature learning.

#### 5.8.2 Equivariant Representations for Robot Manipulation

Prior works have studied equivariant features for robot manipulation [6, 7, 8, 9, 10]. But these works have a series of limitations: (1) they are designed to handle only pick-and-place-like tasks, and are not capable of learning harder tasks involving articulated and deformable objects; (2) they are mostly based on open-loop policies or hand-designed pick-and-place primitives, so their framework does not support closed-loop feedback, which is important for scenarios that involve deformable and articulated objects, where the policy needs to react to changes in object poses throughout task completion; (3) they only handle equivalences in translation and rotation and did not consider scale equivariance. Compared to prior works, our work learns a robot manipulation policy that supports closed-loop feedback, can handle deformable and articulated objects, and is equivariant to objects with different uniform scaling by construction.

#### 5.8.3 3D Representations for Deformable and Articulated Object Manipulation

The problem of 3D manipulation of deformable and articulated objects has been studied by a number of prior works, including FlingBot [34], GarmentNets [35], FabricFlowNet [36], and ACID [37]. However, these works often are very focused on specific manipulation problems. For example, FlingBot focuses on the task of flinging only, FabricFlowNet is designed for pick-and-place tasks on cloths, and ACID is designed to manipulate volumetric deformable objects. We design a framework that is capable of learning a variety of 3D manipulation tasks, including ones that involve deformable and articulated objects.

