# OpenReview forum: "EquivAct: SIM(3)-Equivariant Visuomotor Policies beyond Rigid Object Manipulation"
_robot-learning.org/CoRL/2023/Workshop/OOD — OOD Workshop @ CoRL 2023_

### Official Review · Reviewer_beHc · 2023-10-14
**Good work**

**Rating:** 8
**Confidence:** 4

**Review:**

The paper proposes leveraging SIM(3)-equivariant network to achieve OOD generalization to object translations, rotations, and scaling in deformable manipulation tasks. The network works with point cloud input, and thus compatible with deformable object settings.

The overall approach is simple yet effective. The simulation results clearly indicate the method helps OOD generalization compared to the baselines. The real experiments also use a relatively small number of human demonstrations (20).

A few possible improvements:
- It would be better to write down the contrastive loss formally at the end of Sec. 2.2.
- Include quantitative comparison to baselines for the real experiments
- Include sensitivity analysis on the number of demonstrations. I am a bit concerned that using point cloud information would require more demonstration data compared to 2D input, for example. A discussion on this would be great.

---

### Official Review · Reviewer_3Npp · 2023-10-16
**An approach to learning a visuomotor policy with OOD generalization**

**Rating:** 7
**Confidence:** 4

**Review:**

Herein, the authors address the challenge of learning a visuomotor policy from few examples and generalize zero-shot to scenarios unseen during training. Their proposed SIM(3) equivariant encoder-decoder architecture consists of representation- and policy-learning phases. The representation phase receives point-cloud inputs of some target class with non-uniform scaling, while the policy-learning phase receives a partial point cloud input and utilizes the (fixed) representation-phase’s output of global and local features; the policy phase assumes access to a few demonstrated trajectories of task by an expert.

The authors compare their method to PointNet +BC and PointContrast +BC (with and without augmented data) in simulation, where PointNet + BC has no representation learning phase, whereas PointContrast + BC does. Figure 3 demonstrates the efficacy of the proposed approach in cloth folding, object covering and box closing over the other approaches. Additionally, the authors successfully demonstrate their method on hardware.

While simulated (and experimental) results are promising, and the writing is technically sound, I have a few comments that could strengthen the paper. First, it would be beneficial to compare your (closed-loop) approach to other approaches that only use open-loop policies, as you claim this is a strength of your method. Likewise, are the PointNet and PointContrast baselines closed-loop policies? I do not believe this was delineated in the paper.

Additionally, I would like to see a more precise characterization of “out-of-distribution” generalization: are you just referring to tasks not seen during training? Without clearly stating the novel scenarios are out-of-distribution, it weakens the claim that your method “guarantee out-of-distribution generalization across object translations, 3D rotations, and scales.” In a similar vein, stating your method guarantees OOD generalization is tenuous.

---

### Decision · Program_Chairs · 2023-10-17

**Decision:**

Accept

**Comment:**

We agree with the reviewers’ assessment that this work is technically sound and will contribute to productive, topical discussions at the 2023 Workshop on OOD Generalization in Robotics. In particular, we appreciate the validation of the method with a physical hardware experiment and the effectiveness of the approach on unseen scenarios. We would also like to stress the reviewers’ comments that the impact of this work (in the context of this workshop) would be improved by highlighting specifically how the considered test scenarios are OOD w.r.t. the training data and nuancing strong claims about OOD generalization guarantees when formal proofs of those claims are not provided. We recommend the authors incorporate the reviewers’ feedback into their camera-ready submission to further improve their manuscript.